# Reassessing the Effects of Dietary Fat on Cardiovascular Disease in China: A Review of the Last Three Decades

**DOI:** 10.3390/nu15194214

**Published:** 2023-09-29

**Authors:** Wei Zeng, Qingzhe Jin, Xingguo Wang

**Affiliations:** 1State Key Lab of Food Science and Resources, School of Food Science and Technology, Jiangnan University, 1800 Lihu Avenue, Wuxi 214122, China; leow.zeng@foxmail.com (W.Z.); jqzwuxi@163.com (Q.J.); 2Key Laboratory of Prevention and Treatment of Cardiovascular and Cerebrovascular Diseases, Ministry of Education, School of Basic Medicine, Gannan Medical University, 1 Hexie Avenue, Ganzhou 341000, China

**Keywords:** cardiovascular diseases, high fat diet, fatty acids, atherosclerosis, risk factors

## Abstract

Cardiovascular disease (CVD) is a leading cause of global mortality, and is considered one of diseases with the most rapid growth rate in China. Numerous studies have indicated a closed relationship between an increased incidence of CVD and dietary factors. Dietary fat is one of the three primary nutrients of consumption; however, high fat dietary in causing CVD has been neglected in some official dietary guidelines. Our present review has analyzed the relationship between dietary fat consumption and CVD in China over the past 30 years (from 1990 to 2019). There is a significant correlation between CVD incidence and mortality for consumption of both vegetable oils and animal fats, per capita consumption, and the relative weight of dietary fat exceeding that of other food ingredients (e.g., salt, fruit, and marine food). For fatty acid species, the proportion of ω6 fatty acid consumption increased, causing a significant increase in the ratios of ω6/ω3 fatty acids, whereas the proportion of monounsaturated fatty acid consumption decreased. Such changes have been considered a characteristic of dietary fat consumption in Chinese residents over the past 30 years, and are closely related to the incidence of CVD. Therefore, we suggest that the government should spread awareness regarding the consumption of dietary fat intake to prevent CVD and related health disorders. The public should be educated to avoid high fat diet and increase the intake of monounsaturated fatty acids and ω3 fatty acids.

## 1. Introduction

Cardiovascular disease (CVD) is a broad term encompassing heart and vascular diseases, and is currently the leading cause of global mortality. According to the World Health Organization (WHO), there were 17.9 million deaths from CVD globally in 2019, accounting for 32% of all deaths worldwide [1]. The incidence and mortality rates of CVD in developed Western countries have plateaued, owing to improvements in medical care and preventative measures, whereas the rates in developing countries, mainly China, have continued to increase rapidly. Over the past 30 years, the CVD mortality rate in China has increased from 204.78 to 322.30 per 100,000 people in 2019 (Figure 1), and the incidence rate has increased from 447.81 to 931.57 per 100,000 people, representing an alarming 108.03% increase and the fastest growth rate among all diseases [2]. In 2018, CVD accounted for more than 40% of all-cause mortality in China, ranking first among all fatal diseases [3]. These statistics highlight that CVD has become the Chinese population’s most profound and rapidly growing disease, posing a severe threat to human health.

The most common causes of CVD incidence and mortality include ischemic heart disease, stroke, and congestive heart failure, accounting for >80% of all CVD cases globally [4]. Some of the risk factors for CVD include genetics, age, smoking, physical inactivity, obesity, hypertension, diabetes, and dietary habits. Notably, some of these factors, such as increased life expectancy, reduced physical activity, heightened life stress, and dietary habits have undergone profound transformations in China [5,6]. Approximately 90% of CVD cases are preventable [7], and it is essential to maintain healthy lifestyle habits to reduce the incidence and mortality of CVD.

Dietary factors play a crucial role among the various risk factors for CVD. Improving unhealthy nutritional habits is a primary approach for preventing CVD, as dietary changes are driven by individuals and are typically easier to implement and follow than other interventions. Establishing healthy dietary habits is a convenient, sustainable, and cost-effective approach. Therefore, most countries periodically release dietary guidelines to guide residents in developing healthy eating habits. The Chinese Nutrition Society published the latest edition of the “Chinese Dietary Guidelines (2022)” and the “Report on Scientific Research of Chinese Dietary Guidelines (2021)” [8,9], which cited a study published in *The Lancet Diabetes & Endocrinology* in 2019 by the Chinese Center for Disease Control and Prevention. This study collected dietary intake data from 1982 to 2012 in China and evaluated mortality risk due to cardiovascular and metabolic diseases and their related nutritional factors. The study found that high sodium intake accounted for 17.3% of the cases contributing to cardiovascular and metabolic diseases; insufficient fruit intake accounted for 11.5%; inadequate omega 3 (ω3) fatty acid such as eicosapentaenoic acid (EPA) and docosahexaenoic acid (DHA) intake from seafood accounted for 9.7%; and low intake of vegetables, nuts, whole grains, polyunsaturated fatty acids (PUFA), and low-fat dairy products were ranked fourth and tenth, respectively [10].

The effects of dietary fat on CVD have long been a topic of interest and debate. For instance, prior to 2000, the Dietary Guidelines for Americans (DGA) restricted dietary fat intake less than 30% of total energy intake (%E). However, upon recognizing the potential hazards associated with low-fat diets, the DGA was revised in 2005 to recommend a range of 20–35%E for fat intake. Nevertheless, in 2015, the Dietary Guidelines Advisory Committee recommended the removal of any restrictions on total fat intake [11]. To prevent the onset and death from CVD, topics are generally focused on the type of fatty acids in the diet, with little attention paid to the total fat intake [12,13]. Dietary fats can be divided into vegetable oils and animal fats. According to data from the US Department of Agriculture and the Food (USDA) and Agriculture Organization of the United Nations (FAO), there has been a significant surge in oil and fat consumption in China over the last three decades, with per capita consumption of vegetable oil and animal fat increasing from 5.74 and 8.27 kg in 1990 to 25.84 and 20.56 kg as of 2019, respectively (Figure 2) [14,15].

Figure 3 depicts the changes in energy supply ratios of three major nutrients and total energy intake among Chinese residents from 1992 to 2017. Over the past three decades, there has been a significant decrease in total energy intake; substantial shifts have also occurred in the energy supply ratios of the three major nutrients. These changes are primarily attributed to the reduction in overall cereal consumption. Specifically, the carbohydrate energy supply ratio decreased from 66.2% to 53.3% by 2017, while the fat energy supply ratio increased from 22% to 34.6%. Meanwhile, the protein energy supply ratio exhibited no significant variation. It is noteworthy that excessive consumption of monosaccharides, such as fructose in sugary beverages and increased consumption of sweet snacks, has emerged as an increasingly serious concern in China [16]. These dietary patterns can lead to elevated postprandial triglycerides (TG) levels, adversely impacting metabolic functions and subsequently increasing the risk of developing chronic diseases such as cardiovascular disease, obesity, and diabetes [17,18].

Furthermore, when considering both plant oils and animal fats, the per capita daily consumption among Chinese residents has reached 127.1 g, accounting for more than 30% of total energy intake. Most national dietary guidelines, including those of the WHO, recommend that adults reduce their total fat intake to <30% of their total energy intake [19,20]. However, in most recent studies, high fat intake was not listed as a risk factor for CVD.

## 2. The Relationship between Dietary Patterns and Cardiovascular Disease (CVD)

Table 1 presents the consumption or intake of the major dietary components in China from 1990 to 2019. With a rapid economic growth in China during this period, there have been significant changes in the dietary preferences of Chinese residents. Consumption of nuts, fresh fruits, fresh dairy products, fish and shellfish, and red meat has increased, whereas consumption or intake of salt, total grains, and fresh vegetables has decreased to varying degrees.

### 2.1. Effects of Major Dietary Factors on CVD

Sodium chloride, the primary component of table salt, is the main source of sodium intake in humans. For over a century, several researchers have conducted extensive epidemiological investigations on the relationship between salt intake and CVD in laboratory settings and clinical trials. High salt intake can lead to increased blood pressure and subsequent CVD [23]. Moreover, high salt intake affects blood pressure and causes endothelial dysfunction, stroke, ventricular hypertrophy and fibrosis, arterial and ventricular sclerosis, myocardial infarction, arrhythmia, and heart failure, among other CVD conditions [6,24,25,26]. According to the Chinese Dietary Survey conducted by the Centers for Disease Control and Prevention in 1992, 2002, 2012, and 2015, the average salt intake per person in China showed a declining trend, from 13.9 g/d in 1990 to 9.3 g/d in 2015, representing a decrease of 33.1%. Although this still exceeded the recommended 5 g per day standard set by the Chinese Nutrition Society, the change in salt intake was significantly negatively correlated with CVD incidence in China (*p* < 0.05). Therefore, it is hypothesized that high salt intake is a key factor contributing to the onset and development of CVD. However, sodium salt intake is not the decisive factor causing the high CVD incidence in China.

Considerable evidence suggests that low fruit intake is positively associated with CVD incidence. A massive epidemiological study involving 512,891 participants was conducted to investigate the relationship between the consumption of fresh fruits and major CVDs, and the results indicated a robust logarithmic, linear dose–response relationship between the consumption of fresh fruits and both the incidence and mortality of CVD. Higher fruit consumption is associated with lower blood pressure and blood glucose levels [27]. Fresh fruit is rich in beneficial ingredients, such as polyphenols, vitamins, and minerals, which have various effects on improving all aspects of the cardiovascular system, including inhibiting platelet aggregation and antioxidant and anti-inflammatory effects, thus preventing and treating CVD effectively by reducing blood pressure and improving blood lipids [28,29]. From 1990 to 2019, China’s per capita consumption of fresh fruit increased from 41.6 to 140.7 g, a remarkable growth rate of 238%. Although the per capita consumption of fresh vegetables has been declining annually, the total consumption of fruits and vegetables has remained unchanged over the past 30 years. Therefore, we suggest that low fruit intake has not been the leading cause of the rapid increase in the incidence and mortality of CVD in China over the past three decades.

The influence of ω3 fatty acids on health and disease dates back to the 1970s, when it was discovered that the Inuit people, who mainly consumed meat, had significantly lower rates of CVD than those in other regions. Subsequent studies concluded that the Inuit diet’s high intake of DHA and EPA was responsible for this outcome [30,31]. Extensive evaluation has been conducted on the relationship between ω3 fatty acid intake and chronic diseases, including CVD. Consuming sufficient quantities of ω3 fatty acids has beneficial effects on multiple conditions. Thus, several authoritative international organizations recommend eating fish at least twice weekly to supplement ω3 fatty acids and reduce CVD risk. As shown in Table 1, fish and shellfish consumption in China has increased by 277.1% over the past 30 years, but this has not decreased CVD incidence or CVD-related deaths.

In addition, according to the China Statistical Yearbook, the per capita consumption of nuts, PUFA, and dairy products, which are closely related to the incidence and mortality of CVD, has doubled over the past 30 years. The consumption of vegetables, whole grains, refined grains, and red meat has shown varying degrees of change in the past 30 years, but has tended to stabilize after 2015 and is insufficient to explain the rapid increase in CVD incidence and mortality.

In conclusion, although these 10 factors may contribute to the incidence and mortality of CVD, we unanimously believe that they are not the main reasons for the rapid increase in CVD incidence and mortality.

### 2.2. Contribution of Fat Consumption to CVD

Figure 4a,b shows the linear relationship between vegetable oil and animal fat consumption in China from 1990 to 2019 with the incidence and mortality of CVD. The graph indicates a remarkably robust linear correlation (r = 0.9900, 0.9899) between the consumption of dietary fats (especially vegetable oil) in China and the CVD incidence and mortality over the past 30 years. We also built a random forest machine learning model (Figure 4c,d) based on the consumption of dietary components and the incidence and mortality rates of CVD, and ranked the importance of 12 major food consumption variables. The results also indicated that dietary fat consumption considerably influenced the incidence and mortality of CVD. However, the correlation between CVD and fat intake remains controversial in the medical community, and most studies have indicated that the risk of CVD is not related to total fat intake [32,33]. Nevertheless, based on the data on dietary fat consumption in China over the past 30 years, we believe that there is a highly significant correlation (*p* < 0.0001) between excessive fat intake and CVD.

This situation in China is similar to all countries worldwide. According to the global life expectancy data released by the WHO in December 2020, Japan, Switzerland, and South Korea, which have low per capita consumption of edible oil, ranked among the top three [34], with annual vegetable oil consumptions of 17.7, 17.4, and 19.6 kg, respectively. Japan and South Korea, located in East Asia, have had cancer mortality rates surpassing CVD mortality rates in all-cause mortality since 1996 and 2001, respectively. The countries with the highest CVD incidence rates are concentrated in North America and Western Europe and, coincidentally, these countries also have the highest consumption of edible oils worldwide.

Although fat consumption cannot be equated with intake, per capita consumption and intake are positively correlated in most countries and regions. Epidemiological studies have directly demonstrated a positive correlation between high-fat diets and CVD. A meta-analysis of 27 randomized controlled trials found that reducing dietary fat intake had little effect on all-cause mortality, but could reduce CVD mortality by 9% and CVD incidence by 16%. Moreover, the study also pointed out that as few follow-up studies last more than 2 years, reducing dietary fat intake may have a more meaningful and positive effect on CVD [35]. Another randomized controlled feeding trial involving 217 healthy young adults over 6 months indicated that a high-fat diet was associated with unfavorable changes in gut microbiota, fecal metabolomic profiles, and plasma pro-inflammatory markers, and increased the risk of CVD [36]. In addition, two low-fat diet surveys of postmenopausal women demonstrated that reducing dietary fat intake reduced the incidence and mortality of breast cancer, coronary heart disease, and diabetes as carbohydrate, vegetable, fruit, and grain intake increased with no adverse effects [37,38].

For random forest analysis conditions, a thousand trees were built using R package randomForest (version 4.7–1.1), using data based on the China Statistical Yearbook (1990 to 2019), USDA, and FAO.

IncMSE, the average decrease in prediction accuracy, refers to the increase in prediction error relative to the original error when a variable is randomly permuted and the random forest model is re-estimated. The larger the IncMSE value, the more important the variable is considered to be.

IncNodePurity, the average decrease in node impurity, refers to the impact of a variable on the impurity of decision tree nodes. The larger the IncNodePurity value, the more important the variable is considered to be.

Unfortunately, because altering fat intake in the diet can change other energy sources, micronutrients, dietary fiber, and other nutritional components, it is difficult to establish a reliable and precise correlation between dietary fat and CVD. Thus, there is currently limited literature on the relationship between fat intake and CVD, particularly randomized controlled trials on fat and CVD.

Despite this, high-fat diets significantly increase systolic blood pressure, cholesterol and blood glucose levels, and body mass index, all of which are established risk factors for CVD. An indirect positive correlation between high-fat diets and CVD has also been established. A prospective epidemiological study involving 125,287 participants from 18 countries in North America, South America, Europe, Africa, and Asia found that total fat intake was associated with higher systolic blood pressure. Furthermore, fat intake is associated with elevated total cholesterol (TC) and low-density lipoprotein (LDL) cholesterol levels [39]. A randomized controlled trial and meta-analysis involving 2106 participants from 20 studies demonstrated that in overweight or obese individuals without metabolic disorders, low-fat dietary interventions resulted in varying degrees of TG, TC, LDL, and high-density lipoprotein level reduction [40]. Although there is a consensus on the adverse effects of high-fat diets on the health indicators mentioned above, the relationship between dietary fat intake and CVD remains inconclusive, owing to the potential confounding effects of other nutrients, micronutrients, and dietary fiber. For instance, in 2015, the DGA in the United States recommended the removal of an upper limit for total fat intake. However, the Dietary Reference Intake for total fat intake (20–35%E) established by the National Academy of Medicine in the United States has remained unchanged since 2002. The WHO, expressing concerns about the impact of dietary fat on obesity, has continued to uphold stricter limitations on total fat intake (≤30%E) [41]. Thus, more rigorous randomized controlled trials and comprehensive meta-analyses are required to better understand the relationship between dietary fat and CVD.

## 3. Why Does Excessive Dietary Fat Lead to CVD?

CVD encompasses a wide range of pathologies, and its underlying mechanisms are complex and multifactorial, making it challenging to explain its pathogenesis using a single or a few processes. Nevertheless, most CVDs share a standard pathological change that involves the development of atherosclerosis (AS) and thrombosis. AS is considered the fundamental pathological alteration underlying most CVDs [42]. As early as 1856, German pathologist Rudolf Virchow described lipid accumulation in the arterial wall [43]. American physiologist Ancel Benjamin Keys observed that CVD incidence was high in American corporate executives and Europeans after World War I; it sharply declined during World War II due to reduced food supplies. He proposed the “diet-lipid-heart disease hypothesis”, postulating that CVDs are caused by a high dietary fat intake, which increases blood lipid levels [44]. In turn, reducing blood lipid levels can decrease the incidence of CVD events. However, the pathogenesis of AS remains unclear and complex. Various hypotheses, including oxidative stress, injury response, lipid infiltration, inflammation, and endothelial dysfunction, have been proposed to explain the pathogenesis of AS, but none of them can fully account for it [45]. Nevertheless, abnormal lipid metabolism, endothelial cell dysfunction, inflammatory response, and vascular smooth muscle proliferation are involved in the pathological process of AS (Figure 5) [45,46].

### 3.1. High-Fat Diet Leads to Obesity

A high-fat diet can trigger CVD through multiple mechanisms. First, it causes an imbalance between energy intake and expenditure, leading to increased TG levels and accumulation of white adipose tissue, which can result in obesity. Prospective cohort studies have demonstrated that a high-fat diet in Chinese residents is significantly associated with body weight, body mass index, and obesity [47]. Clinical and epidemiological evidence also indicates that obesity directly or indirectly increases the incidence and mortality of CVD and leads to structural and functional changes in the cardiovascular system to adapt to excess body weight [48]. In addition, under sustained overconsumption of fat, adipocytes are prone to functional disorders, producing and releasing large amounts of pro-inflammatory adipokines and regulating the cellular biology of the heart and blood vessels through endocrine/paracrine signaling pathways, thus inducing the occurrence and progression of CVD [49,50]. Obesity can also indirectly lead to CVD development through diseases, such as insulin resistance, hypertension, and glucose metabolism disorders. For example, obesity-induced insulin resistance can increase lipid accumulation in cardiac myocytes, produce reactive oxygen species and cell apoptosis, and increase oxidative stress responses that damage endothelial cells [51].

### 3.2. High-Fat Diet Leads to Abnormal Lipid Metabolism

A long-term high-fat diet can lead to abnormal lipid metabolism, and blood lipid abnormalities are the main risk factors for CVD [52]. According to a national survey in China in 2012, the incidence rate of blood lipid abnormalities in Chinese adults was as high as 40.4%, significantly higher than that reported in 2002 (18.6%) [53]. The main component of dietary fat is TG, whereas animal fat contains considerable cholesterol. When the levels of TG and cholesterol in the blood are remarkably high, high-density lipoprotein levels decrease, and LDL levels increase accordingly. Lipid metabolism and LDL oxidation modifications play crucial roles in AS development. Lipid metabolism can be divided into exogenous and endogenous. In exogenous lipid metabolism, chylomicrons containing apolipoprotein B-48, C-II, and E are synthesized and secreted by the intestine, and after hydrolysis by lipoprotein lipase, they form chylomicron remnants that are absorbed by the liver via apolipoprotein E. Endogenous lipid metabolism begins with synthesizing very LDL (VLDL) rich in TG. VLDL transports TG to the adipose tissue and partially metabolizes VLDL remnants from LDL [54]. LDL is a carrier that transports cholesterol into the peripheral tissues. Cells recognize and uptake LDL by binding to the N-terminal subunit of apolipoprotein B-100 through LDL receptors.

The concentration of LDL in the blood depends mainly on the uptake of LDL by liver cells via the LDL receptors [55]. In arterial vessels, LDL can penetrate the subendothelial layer and accumulate in the vascular intima by binding to glycosaminoglycans in the extracellular matrix via its surface apolipoprotein B-100 [56]. The deposition of LDL in the vascular wall is considered the first step in AS development [57,58]. Subsequently, LDL residing in the sub-endothelium is oxidatively modified into oxidized LDL (ox-LDL) by reactive oxygen species, such as hypochlorite, phenolic free radicals, and peroxynitrite, induced by inflammation [59,60]. Modification of LDL oxidation can also be mediated by myeloperoxidase and lipoxygenase, which are produced by endothelial cells and macrophages. Oxidative modification of LDL changes the receptors that mediate recognition and endocytosis of LDL from LDL receptors to scavenger receptors. The uptake of LDL by liver cells via LDL receptors can downregulate cholesterol synthesis and maintain cholesterol metabolism balance; however, the uptake of ox-LDL by scavenger receptors cannot downregulate cholesterol synthesis, leading to the accumulation of cholesterol produced by the body and cholesterol derived from dietary animal fats in peripheral cells [61,62]. In addition, ox-LDL can activate various cells, including macrophages, dendritic cells, T cells, endothelial cells, and smooth muscle cells, to induce inflammatory reactions, plaque formation, and AS development [63,64,65].

Therefore, reducing TG, cholesterol, and LDL levels is vital to preventing AS development. Several epidemiological studies and meta-analyses have demonstrated the involvement of cholesterol, TG, and lipoproteins in the pathological process of AS. Elevated blood levels of TG, TC, and LDL significantly increase the risk of developing AS and are linearly related to CVD risk, thus representing an independent risk factor for CVD [66,67,68].

A high intake of processed meat, unprocessed red meat, and poultry (non-fish) is associated with an increased risk of CVD in adults in the United States [69]. Another well-known long-term medical study of Japanese and Japanese American populations found that under the same genetic conditions, long-term adoption of a Westernized diet led to the production of high blood lipid levels and further increased the thickness of the intima-media layer of the carotid artery, accelerating the progression of AS [70].

### 3.3. High-Fat Diet Leads to Microbial Dysbiosis in the Gut

The effect of diet on the gut microbiota is a recent research hotspot. Studies have revealed a close relationship between high-fat diets and alterations in gut microbial abundance, diversity, and richness, leading to an imbalance in the gut microbial ecosystem characterized by an increase in the *Bacteroidetes* and a decrease in the *Firmicutes*, resulting in the dysregulation of lipid metabolism [71,72,73]. A high-fat Western diet also alters the physiological function of the intestinal epithelium, promotes the proliferation of *Escherichia coli* and other Enterobacteriaceae, and enhances the breakdown of choline, which is converted by gut bacteria to trimethylamine (TMA). TMA is absorbed in the gut and oxidized in the liver to TMA-N-oxide, which promotes AS and increases the risk of all-cause mortality [74].

Meanwhile, a high intake of saturated fatty acids (SFAs) has been associated with an increase in *Blautia* bacteria, with alterations in these microorganisms correlate with unfavorable metabolic outcomes, including insulin resistance, higher BMI, and increased waist circumference [75]. Potential mechanisms are thought to be linked to short-chain fatty acids (SCFAs), primarily including acetate (C2), propionate (C3), and butyrate (C4) [76]. SCFAs play a critical role in maintaining human health by acting as signaling molecules on intestinal cells or cells in other tissues, and exerting anti-inflammatory, lipid-lowering, and blood pressure-regulating functions, ultimately reducing inflammation and arterial stiffness [77,78]. Gut microbiota serve as their primary producers, and high SFA diets may diminish SCFA production. In contrast, ω3 fatty acids and dietary fiber are more readily utilized by microbiota to produce SCFAs, contributing to cardiovascular health [79,80].

## 4. The Relationship between Fatty Acid Composition and CVD

Figure 2 also illustrates the consumption of various types of fat and oil in China over the past 30 years. Not only did the total consumption of fats and oils significantly increase, but there has also been a substantial change in the types of fats and oils consumed. In 1990, Chinese residents’ primary vegetable oils were rapeseed, palm, and soybean, accounting for 38.8%, 18.2%, and 16.1% of the total vegetable oil consumption, respectively. Subsequently, with a significant increase in soybean imports, the proportions of these three oils were 21.2%, 10.6%, and 44.5% by 2019, respectively. Regarding animal food consumption, pork, with an average fat content of 34%, is the main type of meat consumed by Chinese residents, accounting for approximately half of the global consumption in the past decade [81]. Although the total pork consumption has increased, its proportion decreased from 79.4% in 1990 to 65.7% in 2019. Correspondingly, the proportions of poultry, beef, eggs, and fishery products have increased, indicating a trend toward the diversified consumption of animal-based foods in China.

### 4.1. Migration of Fatty Acid Consumption in China in the Past 30 Years

Fatty acids play a crucial role in human physiological processes and can be categorized as SFAs and unsaturated fatty acids (UFAs) based on the degree of carbon chain saturation. UFAs can be further divided into monounsaturated fatty acids (MUFAs) and PUFAs based on the number and position of double bonds in the carbon chain, with linoleic acid (LA) and alpha-linolenic acid (ALA) being the most common types of ω6 and ω3 PUFAs, respectively. Different dietary fats have distinct fatty acid compositions, such as high levels of LA in soybean oil, sunflower oil, and cottonseed oil, and high levels of oleic acid in peanut oil, olive oil, and rapeseed oil. Meanwhile, animal fats, palm oil, and coconut oil contain relatively high amounts of SFA.

Figure 6 depicts the changes in the fatty acid composition of the edible oils and fats consumed in China from 1990 to 2019. The figure shows that there has been little change in the proportion of SFAs and MUFAs in animal fat consumed by Chinese residents over the past 30 years, accounting for approximately 40% and 45%, respectively, whereas the ratio of ω6 PUFA to ω3 PUFA is approximately 12:1. However, the fatty acid composition of vegetable oils changed significantly. In 1990, rapeseed oil, which is high in oleic acid, accounted for approximately one-third of the vegetable oil consumption market in China, with MUFA consumption accounting for 41.2%. However, with the increase in the supply of oilseeds, such as soybean, sunflower, and corn, which are rich in LA, ω6 fatty acids have increased from 32.8% to 40.9% over the past 30 years, whereas the proportion of ω9 fatty acids decreased to 34.8% in 2019. The proportions of ω3 fatty acids and SFAs also reduced slightly.

### 4.2. Effect of Fatty Acid Composition on CVD

The relationship between fatty acids and CVD incidence has been extensively studied, and the scientific community has reached a consensus that SFAs and trans-fatty acids are risk factors for CVD [83,84]. Robust evidence has demonstrated a strong causal relationship among SFA intake, blood cholesterol levels, and CVD [85]. Replacing SFAs with UFAs can significantly reduce the risk of CVD [86]. Therefore, in 2019, the WHO recommended reducing the dietary intake of SFAs to <10% of the total energy consumption and increasing the intake of UFAs [19]. In China, although the proportion of SFAs in edible oils has decreased slightly over the past 30 years, the absolute intake of SFAs has hardly changed owing to the overall increase in fat consumption, and it remains lower than that in most Western countries [87,88]. In addition, according to the dietary habits of Chinese residents, the intake of *trans* fatty acids is minimal among the Chinese. As SFAs and *trans* fatty acids are risk factors for CVD, the rapid increase in CVD incidence in China is not related to the current levels of SFA and trans fatty acid consumption.

The significant decrease in the proportion of MUFAs is a prominent feature of China’s dietary fat consumption over the past 30 years, and oleic acid is virtually the sole source of MUFAs in the Chinese diet. The Mediterranean diet, rich in olive oil, is considered one of the healthiest diets globally. Increasing evidence suggests that oleic acid has multiple positive effects on human health and disease, and a high dietary intake of oleic acid in vegetable oils can prevent and alleviate the incidence and mortality of CVD [89,90]. The consumption of oleic acid-rich oil can improve endothelial dysfunction, reduce blood pressure, improve insulin resistance, and exhibit anti-inflammatory and antioxidant effects. Additionally, oils rich in oleic acid can lower total and LDL levels and control AS progression [91]. In 2018, the US Food and Drug Administration issued a statement indicating that replacing oils high in SFA with oils containing >70% oleic acid could reduce the risk of coronary heart disease [92].

In addition to MUFAs, several animal experiments, population studies, and meta-analyses have demonstrated that replacing SFAs with PUFAs can significantly reduce CVD incidence [93,94,95]. This conclusion has also become a part of the dietary guidelines of several countries. Although the focus of these studies is to demonstrate a reduction in CVD events with lowered SFA intake, several studies and media outlets tend to emphasize the benefits of PUFAs. Current evidence supports the varying effects of different fatty acids on CVD risk, which largely depends on the type of fatty acid being compared or replaced [13]. For instance, ω6 fatty acids undoubtedly have a more beneficial effect on CVD than excessive SFAs, but they may have potentially more harmful effects compared with ω3 fatty acids [96].

Dietary fats predominantly contain PUFAs in LA and ALA, which belong to the ω6 and ω3 fatty acid series. These are essential fatty acids, as mammals lack the Δ 12 and Δ 15 desaturase enzymes required for their synthesis [97]. LA and ω6 fatty acids are precursors of arachidonic acid (ARA), which produces several eicosanoids that control various bodily functions, particularly those related to inflammation, and are generally considered pro-inflammatory [98,99]. Conversely, ω3 fatty acids are known to be anti-inflammatory, with abundant evidence showing that EPA and DHA significantly reduce inflammatory markers, such as C-reactive protein, tumor necrosis factor, interleukin-1β, and interleukin-6 [100,101]. LA and ALA undergo a series of biochemical reactions upon ingestion to generate pro-inflammatory ARA, anti-inflammatory EPA, and DHA. Multiple desaturases and elongating enzymes participate in these reactions, and high levels of ALA competitively inhibit the conversion of LA to ARA [102]. AS is a chronic inflammatory disease [103]; therefore, the ω3 to ω6 fatty acid ratio is believed to play a critical role in CVD development. Blood coagulation also contributes to AS development [104], as platelets activated by endothelial cell damage release thromboxane A (TXA), which promotes platelet aggregation and causes vasoconstriction, leading to blood clot formation [105]. In contrast, endothelial cells produce prostacyclin (PGI), which inhibits blood clot formation. The balance between TXA and PGI activity determines the likelihood of clot formation [106,107]. ARA is the sole precursor to procoagulant TXA, and both ARA and EPA can produce the anticoagulant PGI [108]. Although the conversion of ALA to EPA in the human body is limited, increasing the dietary intake of ω3 fatty acids while reducing ω6 fatty acid intake is essential in reducing the risk of CVD (Figure 7).

Both dietary surveys of Chinese residents and assessments of omega-3 fatty acid concentrations in the blood of the Chinese population consistently indicate inadequate ω3 fatty acid intake, with rural residents exhibiting significantly lower levels than their urban counterparts [8,10,109]. With the development of modern agriculture, China’s primary consumption of vegetable oils has shifted from rapeseed oil, with higher levels of ω9 and ω3 fatty acids, to soybean oil, with a higher concentration of ω6 fatty acids, over the past 30 years. In the realm of animal fat consumption, the consumption of milk and red meat has witnessed substantial increases of 515% and 68%, respectively. Notably, red meat, particularly pork, accounts for the majority of animal fat intake and is abundant in saturated fatty acids. Reducing the intake of animal-derived fats can significantly alleviate the burden of saturated fatty acids and choline. In light of these transformations, the dietary ratio of ω6 to ω3 fatty acids among Chinese residents has risen significantly, from 5.8 in 1990 to 7.9 in 2019, marking a substantial increase of 36.2%. A study previous suggested that ω6 to ω3 fatty acid intake ratio was approximately 1:1 while establishing the human genetic profile in the Paleolithic era. However, in current Western diets, this ratio has reached a staggering 15.0:1 to 16.7:1. Reducing this ratio to below 4:1 can result in a 70% reduction in CVD mortality [110].

Therefore, we recommend diversifying cooking oils at the household consumption level, consciously increasing the intake of oils rich in ω3 fatty acids such as flaxseed oil, walnut oil, and rapeseed oil. In terms of animal-based foods, choosing low-fat dairy products, reducing red meat consumption, and incorporating an appropriate increase in the consumption of seafood, including fish, can help adjust the ratio of ω6 to ω3 fatty acid intake to a more suitable range.

## 5. Conclusions

This review has examined the relationship between dietary fat consumption and CVD in China over the past 30 years (1990–2019). Although most researchers believe that high fat intake is not a risk factor for CVD, our analyses demonstrated a strong positive correlation between total dietary fat consumption and the related incidence and mortality in China. In particular, we confirmed the relationship between changes in the proportions of fatty acids consumed in China and CVD over the past 30 years, and determined that the decrease in MUFA consumption and the increase in the ratio of ω6/ω3 fatty acids may also contribute to the rapid increase in CVD incidence and mortality.

Therefore, we recommend reducing the intake of all types of dietary fat to lower fat calories below the WHO-recommended level of 30% of total energy. We also suggest reducing the intake of pork and SFAs in daily diets and appropriately decreasing the intake of ω6 fatty acids, while increasing the consumption of MUFAs and ω3 fatty acids, maintaining a higher level of MUFAs and a lower ω6/ω3 ratio.

Although this review only provided ecological analysis and cannot establish causal relationships, we believe that high consumption and types of dietary fat should be classified as risk factors for CVD. Additionally, given the significant controversies surrounding the relationship between dietary fat and the incidence and mortality of CVD, this scientific question requires more systematic study and investigation, and is a future research topic.

## Figures and Tables

**Figure 1 nutrients-15-04214-f001:**
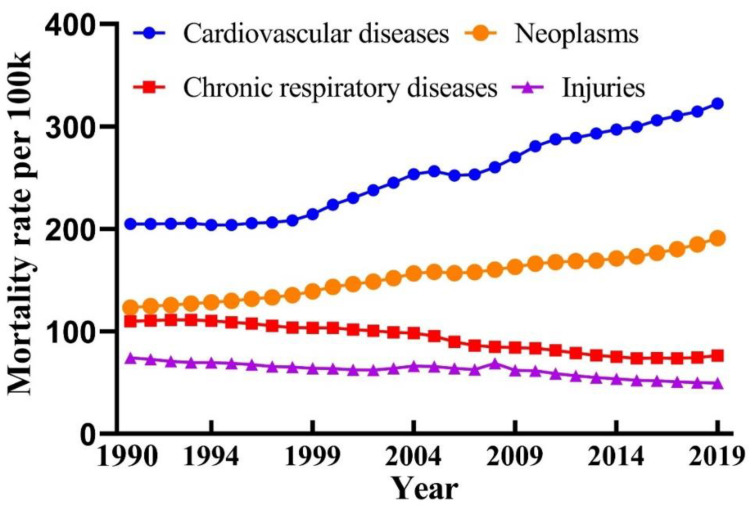
Mortality rate of major diseases in China from 1990 to 2019 [2].

**Figure 2 nutrients-15-04214-f002:**
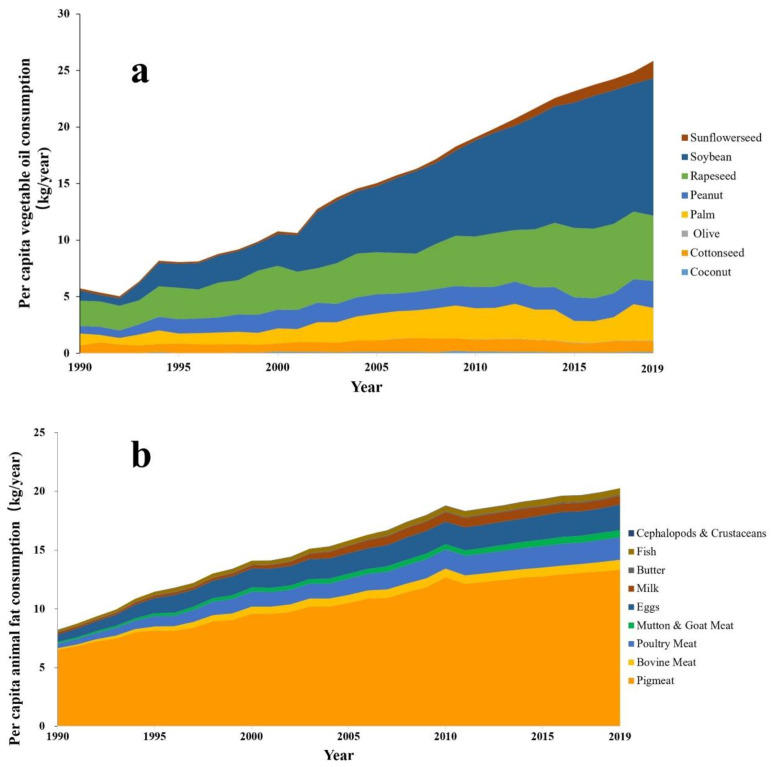
Migration of consumption of vegetable oil (**a**) and animal fat (**b**) in China (1990 to 2019) [14,15].

**Figure 3 nutrients-15-04214-f003:**
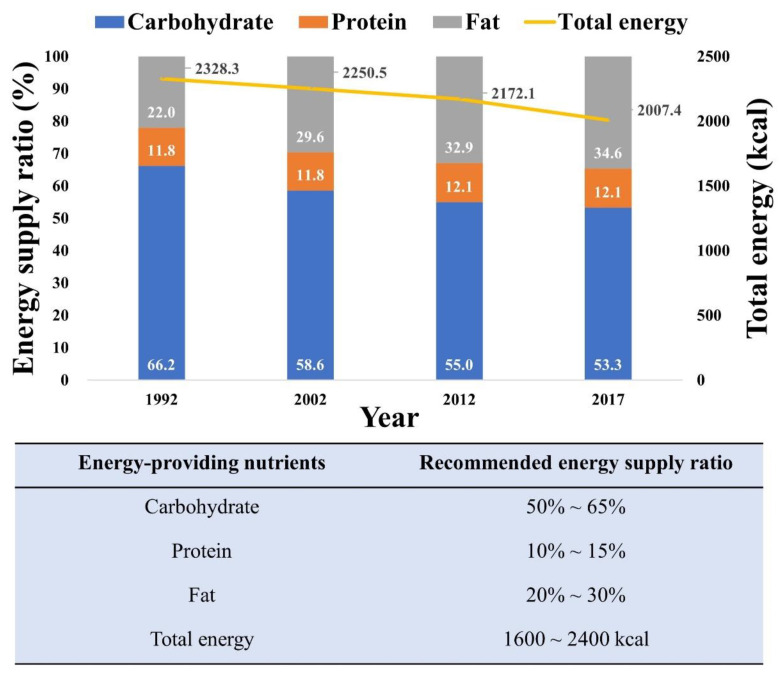
Energy supply ratio of the 3 major nutrients and total energy intake for Chinese residents from 1992 to 2017 [8].

**Figure 4 nutrients-15-04214-f004:**
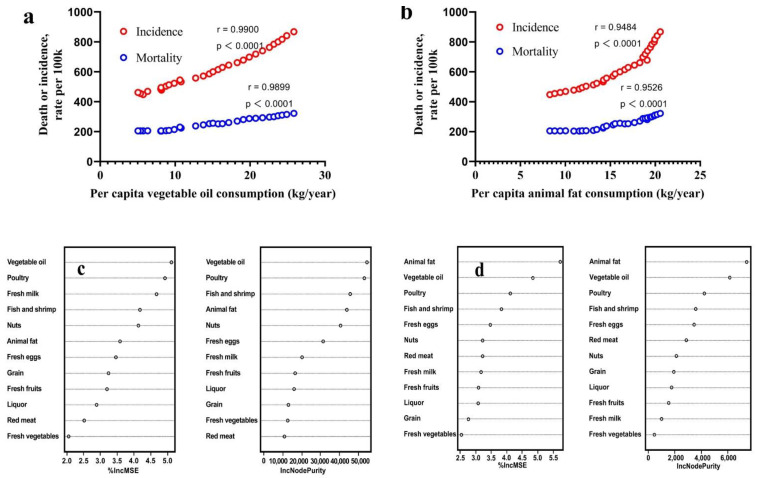
The relationship between per capita dietary fat consumption and cardiovascular diseases in China from 1990 to 2019. (**a**,**b**) the linear relationship between per capita vegetable oil (**a**) and animal fat (**b**) consumption and cardiovascular diseases by Pearson test. (**c**,**d**) the importance ranking based on random forest influencing factors of cardiovascular diseases incidence (**c**) and mortality (**d**).

**Figure 5 nutrients-15-04214-f005:**
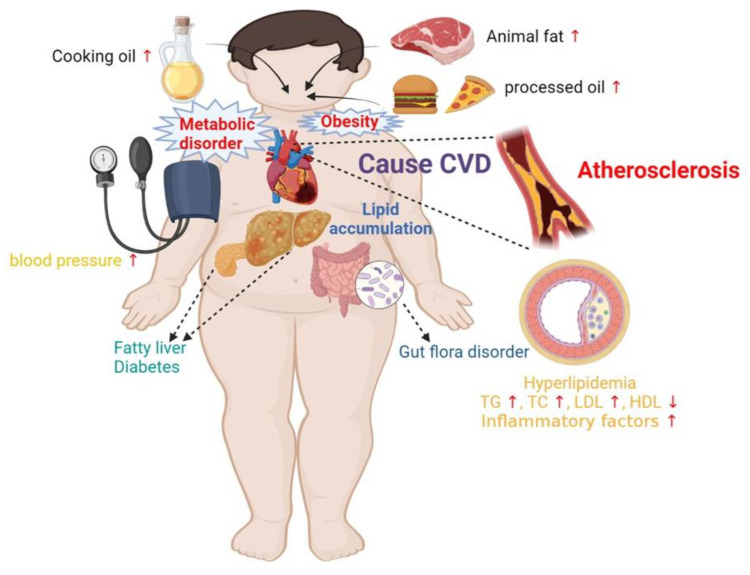
Potential mechanism of high-fat diet-induced cardiovascular diseases. CVD, cardiovascular diseases; TG, triglyceride; TC, total cholesterol; LDL, low-density lipoprotein cholesterol; HDL, high-density lipoprotein cholesterol.

**Figure 6 nutrients-15-04214-f006:**
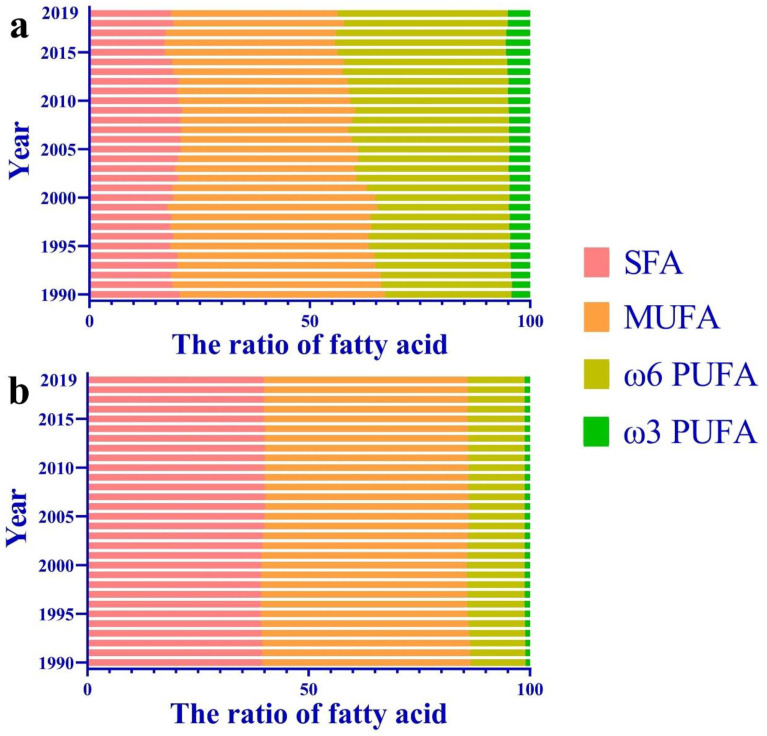
Fatty acid consumption pattern of vegetable oil (**a**) and animal fat (**b**) in China (1990 to 2019). Fatty acid composition calculations by standard food composition tables [82]. SFA, saturated fatty acids; MUFA, monounsaturated fatty acid; PUFA, polyunsaturated fatty acid.

**Figure 7 nutrients-15-04214-f007:**
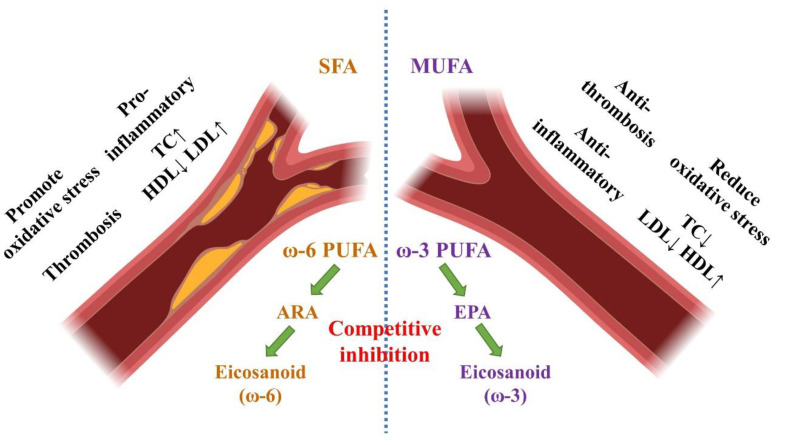
Metabolism of SFA, MUFA, ω6 and ω3 fatty acid link to CVD. SFA, saturated fatty acids; MUFA, monounsaturated fatty acid; PUFA, polyunsaturated fatty acid; ARA, arachidonic acid; EPA, eicosapentaenoic acid; TC, total cholesterol; LDL, low-density lipoprotein cholesterol; HDL, high-density lipoprotein cholesterol.

**Table 1 nutrients-15-04214-t001:** Migration of consumption of main dietary ingredients in China (1990 to 2019) (g/d).

Year	1990	1992	1995	2000	2002	2005	2010	2012	2015	2019	Rate of Change (%)
Salt (g) *	\	13.9	\	\	12.0	\	\	10.5	9.3	\	−33.1%
Nuts (g) **	2.3	\	2.7	4.6	4.3	4.8	5.4	6.0	8.5	10.3	343.8%
Total cereals (g) **	623.0	\	580.5	518.9	478.7	416.9	360.4	326.9	368.5	356.5	−42.8%
Fresh vegetables (g) **	370.5	\	296.1	300.4	309.3	299.4	286.8	271.9	259.9	260.9	−29.6%
Fresh fruits (g) **	41.6	\	61.1	89.0	91.9	93.6	101.1	110.4	110.9	140.7	238.0%
Fresh milk (g) **	5.6	\	4.8	11.7	18.8	25.6	24.0	27.0	33.1	34.3	515.4%
Fish and shrimp (g) **	9.9	\	13.3	18.5	18.8	22.5	27.9	28.8	30.6	37.2	277.1%
Red meat (g) **	38.6	\	37.6	45.1	49.7	54.8	55.2	57.2	63.1	64.8	68.0%

The rate of change refers to the percentage increase or decrease in per capita consumption from 1990 to 2019. *, Data sourced from 2016 and 2022 Chinese Dietary Guidelines [8,21]; **, Data sourced from China Statistical Yearbook (1990 to 2019) [22].

## Data Availability

Data availability is not applicable to this article as no new data were created or analyzed in this study.

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
