# Peer review of "Reassessing the Effects of Dietary Fat on Cardiovascular Disease in China: A Review of the Last Three Decades"

_nutrients, 2023, doi:10.3390/nu15194214_

Round 1

Reviewer 1 Report

This review by Wei Zeng et al. is targeted to evaluate the effects of consumption of fat and quality of fat, i.e. fatty acid distribution especially ratios of ω6/ω3 fatty acids on CVD incidence and mortality in China during the last 30 years. The structure and flow of the review is clear and the conclusions made from the fat data are relevant. It seems that there has been a drastic change in the ratio of ω6/ω3 fatty acids during these years due to nutritional changes where rapeseed consumption has decreased and soybean increased strongly affecting this ratio. SFA levels have been rather stable although pork consumption has somewhat increased. The authors generally suggest that people in China should reduce total dietary fat intake and increase the intake of MUFA and ω3 fatty acids. As a whole the review is very relevant during these days when scientific community is struggling with the issue of dietary fat quality vs CVD risk. However, there are some issues that need further discussion.

MAJOR COMMENTS

1. China is geographically large country and therefore the question is how the data presented in this study operates equally in big urban areas and in rural areas? Is there possibility for a bias in data when collecting data?

2. The authors should also include in their rationalization the issues like physical exercise vs sedentary life style and how they have changed during these 30 years. They are very important factors when evaluating risk for CVD.

3. Chinese dietary guidelines were published for years 1982-2012 indicating already that time inadequate ω3 intake. Assumably also the other conclusions were more or less similar compared to the present study?

4. Fig.3; total energy decreased from 2328 kcal (1992) to 2007 kcal (2017). During this time carbohydrate E% supply decreased from 66.2 to 53.3 % whereas fat E% supply increased from 22 to 34.6%. Is total energy drop due to decreased carbohydrate consumption? Typically it has been a serious problem in many countries that consumption of monosaccharides for instance in the form of beverages including high fructose corn syrup. This has serious consequences in the form of obesity, prediabetes, risk for CVD etc. This should be commented.

4. Since the rapeseed consumption has radically decreased and soybean increased during these 30 years the ratio of ω6/ω3 fatty acids is very high, i.e. 12:1. To reach a recommended ratio of <4:1 needs real efforts. Is the plan to go back and increase consumption of rapeseed oil since the cultivation climate should be beneficial for this plant? Or what is the general plan?

5. Chapter 3.3. SFA also affects the microbiome ability to produce beneficial short-chain fatty acids that are protective. Please add this.

6. Consumption of fresh milk has increased 515% and red meat 68% during 1990-2019. Both add significantly the load of SFA as well as choline load. Reduction of these in every day meals should also be an important target in general fat plan.  

Reviewer 2 Report

The manuscript examines the potential role of total fat on cardiovascular diseases (CVDs) in the last 3 decades in China.

Authors claim that total fat is the main nutritional factor explaining the epidemic of CVDs in China. In my view the narrative review sent by authors is of low priority for publication for the following reasons:

- First, contrary to the best available evidence authors suggest that total fat is a public health problem. This claim is not enough rigorous. Indeed, the type of fat consumed is the key nutritional factor to consider. Trans fats and saturated fats in excess are harmful for cardiovascular health. But authors should recognise that there are essential fats that humans need for health purposes; or should recognise that replacing saturated fats with polyunsatured lowers CVD risk. In sum, the arguments included in the narrative review are misleading with the best existing evidence.

- Second, correlations (ecological studies) are a weak source of scientific evidence. Authors neglect this methodological limitation in their narrative review. A more suitable source would be a well-designed cohort study where authors could include covariates to minimise the risk of confounding (among other biases).

- Third, authors should explain why their recommendations disagree with recommendations of dietary guidelines for fat from USA, where the type of fat is considered and less of total 30% of fat is not recommended.

- Fourth, authors do not take into consideration other CVD causal factors that could explain the increase in prevalence of CVD in the last years. For example, physical inactivity, pollution, other non-nutritional factors (additives, etc) in ultra-proccessed diets, more financial stress problems, less sleep time, etc... To blame fat as main culprit is excessively reductionist. 

In sum, the paper is of low scientific quality because different types of fats need consideration in the nutritional debates. 

  •  

Round 2

Reviewer 2 Report

Despite the manuscript still has important methodological limitations to produce a notable contribution to the literature (ecological studies), the manuscript may be accepted in its current version for two reasons:

- Authors now provide adequate responses to the questions raised in the first review. 

- Second, the effects of a high fat diet on the microbiota need to be explored with new studies. Therefore, the main hypothesis of authors that a fat diet is harmful requires attention as another nutritional factor linking Western diets and chronic diseases. In particular, cheaper oil fats used for cooking could explain in part the rapid increase in the prevalence of CVD diseases in China.